# *Protaetia brevitarsis seulensis* Larvae Extract Attenuates Inflammatory Osteoclast Differentiation and Bone Loss

**DOI:** 10.3390/nu17203273

**Published:** 2025-10-17

**Authors:** Hyun Yang, Dong Ryun Gu, Hye Jin Yang, Wei Li, Younghoon Go, Ra-Yeong Choi, In-Woo Kim, Hyunil Ha

**Affiliations:** 1KM Convergence Research Division, Korea Institute of Oriental Medicine, Yuseong-daero 1672, Yuseong-gu, Daejeon 34054, Republic of Korea; hyunyang@kiom.re.kr (H.Y.); mrwonsin@kiom.re.kr (D.R.G.); 2Korean Medicine-Application Center, Korea Institute of Oriental Medicine, Daegu 41062, Republic of Korea; hjyang@kiom.re.kr (H.J.Y.); liwei1986@kiom.re.kr (W.L.); gotra827@kiom.re.kr (Y.G.); 3Department of Agricultural Biology, National Institute of Agricultural Sciences, Rural Development Administration, Wanju 55365, Republic of Korea; fkdud1304@korea.kr (R.-Y.C.); kiw0601@korea.kr (I.-W.K.)

**Keywords:** *Protaetia brevitarsis seulensis*, osteoclast, RANKL, prostaglandin E2, inflammation

## Abstract

**Background/Objectives:** The larvae of *Protaetia brevitarsis seulensis* (PB), an edible insect, exhibit diverse bioactivities, but their effects on inflammatory bone loss remain unclear. We investigated whether a 70% ethanol extract of PB larvae (PBE) suppresses osteoclast differentiation and bone loss under inflammatory conditions. **Methods:** Osteoclast differentiation was assessed in co-cultures of mouse bone marrow cells and osteocytic cells stimulated with interleukin-1 (IL-1). Direct effects on osteoclast precursors were tested in bone marrow–derived macrophages exposed to receptor activator of nuclear factor-κB ligand (RANKL) or tumor necrosis factor-α (TNF-α). Skeletal effects were evaluated in a mouse model of lipopolysaccharide (LPS)-induced bone loss. **Results:** PBE inhibited IL-1–induced osteoclast differentiation in co-culture, reduced osteocytic RANKL expression and prostaglandin E2 (PGE2) production, and dampened early IL-1 signaling. In osteoclast precursors, PBE directly suppressed osteoclastogenesis driven by RANKL or TNF-α. In vivo, PBE attenuated LPS-induced bone loss and blunted the associated increases in bone RANKL and PGE2. **Conclusions:** PBE limits inflammatory osteoclastogenesis by downregulating PGE2 and RANKL production in osteoclast-supporting cells and directly inhibiting osteoclast precursor differentiation, thereby attenuating LPS-induced bone loss. These findings identify PBE as a food-derived candidate for managing inflammation-associated bone loss and support further preclinical and nutritional intervention studies.

## 1. Introduction

Bone is a specialized connective tissue composed of a mineralized matrix and is continuously remodeled through the tightly coupled actions of bone-resorbing osteoclasts and bone-forming osteoblasts. Osteoblasts arise from mesenchymal stem cells, whereas osteoclasts are multinucleated giant cells derived from the monocyte/macrophage lineage [1]. Osteoclast differentiation is governed primarily by macrophage colony-stimulating factor (M-CSF) and receptor activator of nuclear factor-κB ligand (RANKL): M-CSF promotes the survival and proliferation of osteoclast precursors, while RANKL engages its receptor RANK to drive commitment, fusion, and activation [2]. Aberrant RANKL signaling and excessive osteoclast activity contribute to bone-destructive conditions such as postmenopausal osteoporosis, periodontitis, lytic bone metastases, and rheumatoid arthritis [1,3].

Within the bone microenvironment, multiple cell types—including B lymphocytes and mesenchymal stem cell–derived lineages such as osteoblasts and adipocytes, as well as osteocytes—modulate osteoclastogenesis by producing M-CSF, RANKL, and/or osteoprotegerin (OPG) [4,5,6,7]. OPG, a secreted decoy receptor for RANKL, sequesters RANKL to restrain osteoclastogenesis [2], and leucine-rich repeat-containing G protein-coupled receptor 4 (LGR4) can bind RANKL to temper RANK signaling [8].

Inflammation is a potent accelerator of osteoclast differentiation. Pro-inflammatory cytokines, notably interleukin-1 (IL-1) and tumor necrosis factor-α (TNF-α), elevate RANKL and potentiate osteoclastogenesis [9,10,11,12,13], contributing to bone loss in multiple disorders [14,15,16,17]. Lipopolysaccharide (LPS) is widely used to model inflammation-induced osteoclastogenesis and bone loss in rodents [18], where RANKL, prostaglandin E2 (PGE2), and TNF-α are key mediators [5,19,20,21].

Edible insects are nutrient-dense resources increasingly explored for nutraceuticals and functional foods [22]. The white-spotted flower chafer *Protaetia brevitarsis*, including the subspecies *P. brevitarsis seulensis* (PB), is approved in Korea as a general food ingredient. PB larvae contain high protein, free amino acids, fatty acids (notably oleic acid), organic acids (e.g., citric and succinic acids), minerals, and small molecules (e.g., nucleosides) [23,24,25,26]. Of note, in a 70% ethanol extract of PB larvae (PBE), L-tryptophan has been isolated and quantified as a quality-control marker [27].

PB larvae and their extracts exhibit diverse bioactivities [26,28,29,30]. Notably, PBE improved bone loss in ovariectomized mice, inhibited RANKL-induced osteoclastogenesis, and reduced LPS-induced inflammatory responses [31,32]. However, whether PBE mitigates inflammation-driven osteoclastogenesis and the associated bone loss remains unclear. Here, we tested the hypothesis that PBE suppresses inflammatory osteoclastogenesis and protects bone by integrating assays of osteocytic pro-osteoclastogenic signaling and osteoclast precursor differentiation with an in vivo model of LPS-induced inflammatory bone loss.

## 2. Materials and Methods

### 2.1. Preparation of PBE

PBE was prepared from freeze-dried powder of PB larvae using 70% ethanol at 75 °C for 6 h under Good Manufacturing Practice (GMP) conditions, and the crude hydroethanolic extract was concentrated and freeze-dried, as described previously [27]. The process yield was 14.4% ± 2.7% (*w*/*w*; *n* = 3 GMP batches). A single GMP-prepared batch (batch 1 described in [27]) was used for all experiments; batch identity was verified by HPLC-DAD fingerprinting and by quantifying L-tryptophan as a quality-control marker [27]. PBE powder was aliquoted into amber vials, protected from light, and stored at −20 °C; each aliquot was thawed once at the time of use. The maximum storage duration was ≤6 months.

### 2.2. Cytokines and Antibodies

Recombinant mouse IL-1α, mouse TNF-α, and human transforming growth factor-β1 (TGF-β1) were obtained from PeproTech (East Windsor, NJ, USA). LPS (from *Escherichia coli* O111:B4) was purchased from Sigma-Aldrich (St. Louis, MO, USA). Recombinant human M-CSF and RANKL were prepared as previously described [33]. Primary antibodies against phospho-ERK1/2 (Thr202/Tyr204, #9101), ERK1/2 (#9102), phospho-JNK (Thr183/Tyr185, #9251), JNK (#9252), phospho-p38 (Thr180/Tyr182, #9211), p38 (#9212), phospho-Akt (Ser473, #9271), Akt (#9272), IκBα (#9242), and β-actin (#3700) were from Cell Signaling Technology (Danvers, MA, USA).

### 2.3. Osteoclast Differentiation Assays

MLO-Y4 (Kerafast, Boston, MA, USA), a murine osteocyte-like cell line, was used as an in vitro osteocyte model due to its dendritic morphology and expression of osteocyte markers. MLO-Y4 cells were seeded in 96-well plates at 1 × 10^3^ cells/well and cultured in α-MEM supplemented with 10% heat-inactivated FBS and 1% penicillin-streptomycin. The following day, bone marrow was flushed from the femora and tibiae of C57BL/6J mice with phosphate-buffered saline (PBS) using a 3 mL syringe, and red blood cells were lysed with ACK Lysing Buffer (Thermo Fisher Scientific, Waltham, MA, USA). The resulting bone marrow cells (1 × 10^5^ cells/well) were added to the MLO-Y4 cultures to establish co-cultures. Co-cultures were pretreated with PBE or dimethyl sulfoxide (DMSO) for 1 day and then stimulated with IL-1 (10 ng/mL) or IL-1 plus RANKL (50 ng/mL) for 5 days to induce osteoclast differentiation.

Mouse bone marrow–derived macrophages (BMMs; 1 × 10^4^ cells/well), prepared from bone marrow cells by treatment with M-CSF [33], were cultured in 96-well plates with M-CSF (60 ng/mL). BMMs were pretreated with PBE for 24 h and then stimulated with RANKL (50 ng/mL) for 4 days or with TGF-β1 (1 ng/mL) plus TNF-α (20 ng/mL) for 5 days. Cell viability was measured after 2 days using a Cell Counting Kit-8 (Dojindo Molecular Technologies Inc., Rockville, MD, USA).

Osteoclast differentiation was evaluated by total tartrate-resistant acid phosphatase (TRAP) activity and TRAP staining following fixation and permeabilization [34]. Osteoclasts were defined as TRAP-positive multinucleated cells with >3 nuclei and a diameter >100 μm.

### 2.4. Western Blot Analysis

MLO-Y4 cells (2 × 10^5^ cells/well) were maintained in 6-well plates for 24 h, pretreated with or without PBE for 24 h, and subsequently stimulated with IL-1α for the indicated times. Cells were washed twice with ice-cold PBS and lysed in PRO-PREP lysis buffer (iNtRON Biotechnology, Seongnam, Republic of Korea). Equal amounts of protein (15 μg) were separated by SDS-PAGE, transferred to PVDF membranes, and analyzed by Western blotting as previously described [34].

### 2.5. Quantitative Real-Time Polymerase Chain Reaction (qRT-PCR)

Cell seeding and pretreatment conditions were identical to Section 2.4. Total RNA was extracted using the RNeasy Mini Kit (Qiagen, Hilden, Germany). cDNA was synthesized from 2 μg of total RNA using a High-Capacity cDNA Reverse Transcription Kit (Applied Biosystems, Foster City, CA, USA). qRT-PCR was performed using TaqMan Universal Master Mix II and the following TaqMan gene expression assays: Tnfsf11 (Mm00441908_m1), Tnfrsf11b (Mm00435454_m1), Ptgs2 (Mm00478374_m1), Ptges (Mm00452105_m1), Lgr4 (Mm00554385_m1), and 18S rRNA (Hs99999901_s1), on a QuantStudio 6 Flex system (Applied Biosystems). Relative gene expression was calculated by the ΔΔCt method with 18S rRNA as the internal control.

### 2.6. Animal Study

All animal procedures were approved by the Institutional Animal Care and Use Committee of the Korea Institute of Oriental Medicine (IACUC protocol code: 25-067; approval date: 30 June 2025). The study was designed and reported in accordance with the ARRIVE guidelines. Female C57BL/6J mice (8 weeks old) were obtained from Saeronbio Inc. (Uiwang, Republic of Korea) and housed in a specific pathogen-free facility with ad libitum access to food and water. After 2 weeks of acclimatization, mice were randomly assigned to the four groups (*n* = 7/group) using a computer-generated sequence (groups balanced for baseline body weight): sham, LPS, PBE-L (PBE 100 mg/kg/day + LPS), and PBE-H (PBE 300 mg/kg/day + LPS). The primary outcome for the in vivo study was distal femoral trabecular bone volume fraction. An a priori power analysis (G*Power version 3.1.9.7; one-way ANOVA, 4 groups, α = 0.05, power = 0.80; f = 0.63) based on literature for LPS-induced trabecular loss and oral PBE efficacy yielded *N* = 28 (*n* = 7/group) [18,31]. The doses (100 and 300 mg/kg/day) were chosen based on prior efficacy of oral PBE at 100 and 200 mg/kg in protecting against bone loss in ovariectomized mice [31]. Mice received intraperitoneal injections of PBS or LPS (5 mg/kg) on days 0 and 4, and femora were collected on day 8 after a 7 h fast. PBE or distilled water (DW) was administered orally once daily from 1 week before the first LPS injection until sacrifice. Animals were monitored daily with prespecified humane endpoints, none of which were reached. Each mouse was considered an experimental unit. To minimize potential confounding effects, treatment assignments and measurements were randomized, cage positions were interleaved, and handling was rotated among groups. Allocation and animal handling were conducted by personnel aware of group assignments, and outcome assessment and data analysis were performed blinded to group allocation. Left femora were fixed in 10% neutral buffered formalin for micro-computed tomography (μ-CT) analysis, and right femora were stored at −70 °C for protein extraction. Right femora were washed with PBS, the proximal and distal epiphyses were removed, and the remaining femoral segments comprising the metaphysis and diaphysis were homogenized in 400 μL of lysis buffer [50 mM Tris-HCl (pH 7.4), 150 mM NaCl, 0.5% Triton X-100, 1 mM EDTA, and a protease/phosphatase inhibitor cocktail (Thermo Fisher Scientific)]. Lysates were centrifuged at 12,000× *g* for 10 min at 4 °C, and supernatants were collected for protein quantification using a BCA assay (Thermo Fisher Scientific). Samples were used for enzyme-linked immunosorbent assay (ELISA) of RANKL, OPG, and PGE2. No animals were excluded from μ-CT or body/spleen-weight analyses (*n* = 7/group). For ELISAs of bone lysates, a small number of samples were not analyzable due to technical loss during tissue processing. No data were removed based on outcome values.

### 2.7. μ-CT Analysis

Distal femora were scanned using a μ-CT system (SkyScan 1276, Bruker, Kontich, Belgium) at 8 μm isotropic voxel size with a 0.8° rotation step. Images were reconstructed with NRecon (v1.7.42), morphometric parameters were analyzed using CTAn (v1.20.3.0), and three-dimensional visualization was performed with CTvol (v3.3.0r1412). The trabecular bone volume of interest was defined as a 1.75 mm region extending proximally, starting 0.25 mm from the distal femoral growth plate.

### 2.8. ELISA

RANKL levels in MLO-Y4 cell lysates (8 μg total protein) and bone lysates (90 μg), as well as OPG levels in bone lysates (25 μg), were quantified using commercial ELISA kits (R&D Systems, Minneapolis, MN, USA). PGE2 concentrations in bone lysates (6 μg) and MLO-Y4 culture supernatants (0.1 μL) were determined using an ELISA kit (Cayman Chemical, Ann Arbor, MI, USA).

### 2.9. Statistical Analysis

Statistical analyses were performed using GraphPad Prism 9 (GraphPad Software, La Jolla, CA, USA). In vitro data are presented as mean ± SD, and in vivo data as mean ± SEM. Group differences were evaluated by one-way ANOVA with Dunnett’s post hoc test or two-way ANOVA with Sidak’s post hoc test. Data normality was assessed using the Shapiro–Wilk test, and homogeneity of variances using the Brown–Forsythe test. When these assumptions were not met, data were log-transformed or analyzed using the non-parametric Kruskal–Wallis test.

## 3. Results

### 3.1. PBE Inhibits IL-1–Induced Osteoclast Differentiation by Suppressing RANKL Expression

Osteocytic MLO-Y4 cells can support osteoclast differentiation through robust expression of M-CSF and RANKL [35]. The pro-inflammatory cytokine IL-1 promotes osteoclast differentiation by stimulating RANKL expression in osteoclast-supporting cells and contributes to inflammatory bone loss [10,15,17,36]. To investigate the effect of PBE on osteoclast differentiation under inflammatory conditions, mouse bone marrow cells, including osteoclast precursors, were co-cultured with osteoclast-supporting MLO-Y4 cells. In MLO-Y4/bone marrow cell co-cultures, IL-1 markedly increased osteoclast differentiation, as evidenced by elevations in total TRAP activity and the number of TRAP-positive multinucleated cells. PBE inhibited IL-1–induced osteoclastogenesis in a dose-dependent manner (Figure 1A–C).

To determine whether PBE modulates the osteoclastogenic capacity of MLO-Y4 cells, we examined RANKL/OPG expression. A 24 H exposure to IL-1 increased *Tnfsf11* (encoding RANKL) mRNA without affecting *Tnfrsf11b* (encoding OPG) (Figure 2A). At concentrations that robustly suppressed osteoclast formation in co-culture (100–200 µg/mL), PBE reduced *Tnfsf11* expression under both basal and IL-1–stimulated conditions (Figure 2A) and decreased RANKL protein levels (Figure 2B). IL-1 decreased *Lgr4* mRNA, and this change was not altered by PBE (Figure 2A). Given the established role of PGE2 in sustaining IL-1–driven RANKL induction, we quantified PGE2 and its biosynthetic enzymes. Cyclooxygenase-2 (COX-2) and microsomal prostaglandin E synthase-1 (mPGES-1) are induced by inflammatory stimuli, including IL-1 and LPS, and are required for PGE2 upregulation [10,20]. IL-1 markedly elevated PGE2 in culture supernatants, and this increase was attenuated by PBE (Figure 2C), concomitant with suppression of *Ptgs2* (COX-2) and *Ptges* (mPGES-1) transcripts (Figure 2A). PBE also blunted the early induction of *Tnfsf11* at 3–6 h after IL-1 stimulation (Figure 2D).

### 3.2. PBE Attenuates Early IL-1 Signaling in Osteocytic Cells

Given that IL-1 rapidly activates ERK, JNK, and p38 mitogen-activated protein kinases (MAPKs) and Akt, as well as the canonical nuclear factor kappa B (NF-κB) pathway, to induce *Tnfsf11* [37], we assessed early signaling in MLO-Y4 cells. IL-1 increased phosphorylation of ERK, JNK, p38, and Akt and induced IκBα degradation. PBE pretreatment reduced IL-1–induced phosphorylation of ERK, JNK, and Akt, but had no discernible effect on p38 phosphorylation or IκBα degradation (Figure 3). These findings suggest that PBE attenuates early RANKL induction mainly by dampening ERK/JNK/Akt activation.

### 3.3. PBE Directly Suppresses Osteoclast Precursor Differentiation

To test whether reduced osteocytic RANKL fully explains the anti-osteoclastogenic effect of PBE, exogenous RANKL was added to MLO-Y4/bone marrow cell co-cultures. IL-1 plus RANKL further enhanced osteoclastogenesis, yet PBE continued to decrease total TRAP activity and the number of multinucleated osteoclasts (Figure 4A), implying additional, osteocyte-independent actions. In BMM monocultures, PBE significantly reduced RANKL-driven osteoclast formation without reducing viability (Figure 4B), consistent with prior observations in RAW264.7 cells [31]. Because TNF-α can induce osteoclastogenesis independently of the RANKL–RANK axis [38], we also examined an inflammatory model. PBE dose-dependently attenuated TNF-α plus TGF-β1–induced TRAP activity and osteoclast formation, again without reducing viability (Figure 4C). Thus, these findings demonstrate that PBE inhibits osteoclast differentiation both by limiting osteocytic RANKL expression and by directly targeting precursor responses under RANKL-dependent and TNF-α–driven conditions.

### 3.4. PBE Mitigates LPS-Induced Bone Loss In Vivo

We next evaluated whether the cellular effects of PBE translate into skeletal protection in an LPS-induced inflammatory bone loss model [20,21]. Systemic LPS administration (5 mg/kg, intraperitoneal; on days 0 and 4) caused pronounced trabecular bone loss in the distal femur, with significant reductions in bone mineral density (BMD), bone volume per tissue volume (BV/TV), and trabecular number (Tb.N), accompanied by an increase in trabecular separation (Tb.Sp), without changes in trabecular thickness (Tb.Th). Compared with LPS controls, oral PBE (100 or 300 mg/kg/day) attenuated the decreases in BMD, BV/TV, and Tb.N; PBE-L (100 mg/kg/day) reduced the LPS-induced increase in Tb.Sp, whereas PBE-H (300 mg/kg/day) increased Tb.Th (Figure 5A,B). LPS increased spleen weight without altering body weight, and PBE had no effect on either parameter (Figure 5C).

To explore potential mechanisms, femoral lysates were analyzed for osteoclast-related factors. LPS elevated RANKL and PGE2, whereas OPG remained unchanged at the study endpoint. PBE dose-dependently suppressed RANKL and PGE2 without affecting OPG (Figure 6A–C). Collectively, these findings indicate that PBE protects against inflammatory bone loss, at least in part, by dampening osteoclastogenic signaling within bone tissue.

## 4. Discussion

This study demonstrates that PBE suppresses inflammatory osteoclastogenesis through coordinated actions on both osteoclast-supporting cells and osteoclast precursors. In co-culture, PBE reduced IL-1–driven osteoclast formation alongside decreases in *Tnfsf11*/RANKL and PGE2 in MLO-Y4 cells, while *Tnfrsf11b* was unchanged and IL-1–mediated reductions in *Lgr4* were not altered. In precursor monocultures, PBE directly inhibited differentiation elicited by either RANKL or TNF-α. Consistent effects were observed in vivo, where oral PBE attenuated LPS-induced trabecular bone loss and lowered bone tissue RANKL and PGE2 without affecting OPG.

Prior reports indicate that RANKL induction in osteoclast-supporting cells in response to inflammatory stimuli, including IL-1, TNF-α, and LPS, proceeds in two temporal phases: an early, PGE2-independent rise and a delayed, PGE2-dependent sustainment [10,39,40]. In the present study, PBE reduced the early increase in *Tnfsf11* (3–6 h) and decreased PGE2 at 24 h, in parallel with reduced *Ptgs2* and *Ptges* expression. Inflammatory cytokines, including IL-1, augment PGE2 production by enhancing activation and expression of cytosolic phospholipase A2 (cPLA2), COX-2, and mPGES-1 in osteoblast-lineage cells [10,20,21]. Thus, PBE-mediated suppression of *Ptgs2*/*Ptges* provides a mechanistic basis for lower PGE2 and, consequently, diminished RANKL at later time points. Early signaling analyses further support a reduction in IL-1–evoked pathways implicated in rapid RANKL upregulation. IL-1 increased phosphorylation of ERK, JNK, p38, and Akt and induced IκBα degradation in MLO-Y4 cells, in line with previous reports in osteoblast-lineage cells [37]. Pharmacologic inhibition of MAPKs, Akt, or NF-κB suppresses early RANKL induction by IL-1 [37], and MyD88-dependent ERK activation contributes to RANKL expression in response to IL-1, LPS, or diacyl lipopeptide [36]. In our system, PBE attenuated IL-1–induced ERK, JNK, and Akt phosphorylation without detectable effects on p38 phosphorylation or IκBα degradation, consistent with the blunted early *Tnfsf11* response and selective modulation of pathways closely linked to RANKL induction. Given that IL-1–driven MAPK and NF-κB activation proceeds via the MyD88–TRAF6 axis [41], preserved p38 and IκBα responses argue against a broad upstream block. Instead, the pattern suggests selective interference at nodes feeding ERK/JNK and PI3K–Akt, although PBE’s direct molecular targets remain unclear.

Our data also indicate that osteocyte-directed mechanisms alone do not fully account for the anti-osteoclastogenic action of PBE. The persistence of activity in co-culture despite exogenous RANKL, together with direct inhibition of RANKL-driven osteoclastogenesis in BMMs and suppression of TNF-α–driven osteoclast formation, supports precursor-intrinsic targets downstream of RANK and within TNF-α pathways.

In addition to the RANKL decoy receptor OPG, LGR4 also binds RANKL and suppresses canonical RANK signaling to limit osteoclastogenesis; notably, LGR4 expression increases during osteoclast differentiation [8]. Moreover, LGR4 in osteoblast-lineage cells negatively regulates osteoclast differentiation and bone resorption [42]. However, PBE did not elevate *Tnfrsf11b* or rescue IL-1–induced reductions in *Lgr4*, suggesting that these RANKL-antagonizing mechanisms are unlikely to account for its effects in our system. Rather, the precursor-directed actions are consistent with earlier reports that PBE downregulates NFATc1, a crucial transcription factor for osteoclastogenesis, and downstream osteoclastic genes [31], and may extend to TNF-α–dependent programs that can proceed independently of the RANKL–RANK axis [38].

The in vivo LPS model further supports translational relevance. PBE inhibited LPS-induced trabecular bone loss without affecting body weight or LPS-induced splenomegaly. Concomitantly, bone RANKL and PGE2 decreased, whereas OPG was unchanged, paralleling the cellular findings. Systemic LPS is known to induce trabecular bone loss that depends on RANKL, TNF-α, and PGE2; OPG administration or TNFR1 deletion, as well as genetic disruption of PGE2 synthesis/signaling (cPLA2, mPGES-1, or EP4), reduces LPS-induced osteoclastogenesis and bone resorption [19,20,21,40]. Beyond elevating RANKL in supporting cells [10,39,40], PGE2 can directly promote migration and differentiation of osteoclast-lineage cells via EP4, and myeloid EP4 deficiency reduces osteoclast numbers in murine models [43]. Thus, PBE-mediated reduction in bone PGE2 likely contributes to protection by lowering RANKL availability and limiting PGE2-driven effects on osteoclasts and their precursors. Collectively, these in vivo data suggest that decreases in bone PGE2 and RANKL are major contributors to the protective effect of PBE against LPS-induced bone loss. The LPS-induced bone loss model used here reflects an acute systemic inflammatory state, and translation to chronic disorders (e.g., rheumatoid arthritis, postmenopausal osteoporosis) requires caution. Notably, PBE ameliorated bone loss in ovariectomized mice, supporting potential efficacy beyond acute settings [31]. However, rigorous evaluation in chronic inflammatory bone loss models is warranted to define durability, dosing windows, and safety.

PBE prepared with 70% ethanol was used in this study. Because 70% ethanol preferentially extracts mid-polarity constituents, recovery of highly polar macromolecules (e.g., certain polysaccharides) is likely limited. Previous studies indicate that PBE comprises amino acids, fatty acids, organic acids, sugars, and nucleosides [31]; in our preparation, L-tryptophan was an abundant constituent and served as a quality-control marker [27]. Although oral PBE was efficacious in vivo here and in ovariectomized mice [31], formal pharmacokinetic and metabolic stability data for PBE (and its putative actives) are lacking. Microbial metabolites of dietary tryptophan have been implicated in bone regulation [44], but the effect of oral L-tryptophan per se in this context remains to be defined. Defining the major bioactive constituents, their oral bioavailability, metabolic fate, and target engagement will be essential for nutraceutical development.

## 5. Conclusions

PBE decreases inflammatory osteoclastogenesis by reducing RANKL and PGE2 production in osteoclast-supporting cells and by directly limiting osteoclast precursor differentiation under both RANKL- and TNF-α–driven conditions. These cellular actions translate in vivo: PBE attenuated LPS-induced trabecular bone loss and lowered bone RANKL and PGE2 without affecting OPG. Collectively, these findings support PBE as a food-derived candidate for managing inflammation-associated bone loss and motivate evaluation of PBE in chronic inflammatory bone loss models and the identification of active constituents.

## Figures and Tables

**Figure 1 nutrients-17-03273-f001:**
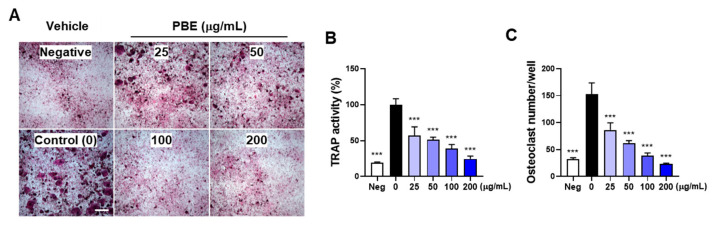
PBE inhibits IL-1–induced osteoclast differentiation in co-cultures. Mouse bone marrow cells were co-cultured with osteocytic MLO-Y4 cells, pretreated with vehicle (0.1% DMSO) or PBE at the indicated concentrations, and then stimulated with PBS (negative control, Neg) or IL-1 (10 ng/mL) for 5 days. After fixation and permeabilization, TRAP staining was performed (**A**); scale bar, 500 μm), and total TRAP activity (**B**) and the number of TRAP-positive multinucleated osteoclasts (**C**) were quantified. Data are presented as mean ± SD (*n* = 3). *** *p* < 0.001 versus IL-1-treated vehicle.

**Figure 2 nutrients-17-03273-f002:**
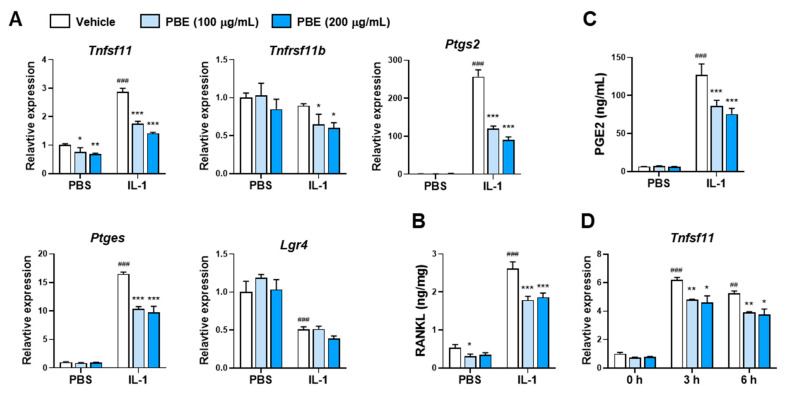
PBE suppresses IL-1–induced RANKL expression and PGE2 production in MLO-Y4 cells. (**A**–**C**) MLO-Y4 cells were pretreated with vehicle or PBE (100 or 200 μg/mL) for 24 h and then stimulated with IL-1 (10 ng/mL) or PBS for 24 h. mRNA levels of *Tnfsf11*, *Tnfrsf11b*, *Ptgs2*, *Ptges*, and *Lgr4* were analyzed by qRT-PCR (**A**). RANKL protein in cell lysates (**B**) and PGE2 in culture supernatants (**C**) were measured by ELISA. (**D**) Time-course analysis of *Tnfsf11* induction: MLO-Y4 cells were pretreated with PBE and stimulated with IL-1 for 3 or 6 h, followed by qRT-PCR. Data are presented as mean ± SD (*n* = 3). ## *p* < 0.01, ### *p* < 0.001 versus control without IL-1; * *p* < 0.05, ** *p* < 0.01, *** *p* < 0.001 versus vehicle.

**Figure 3 nutrients-17-03273-f003:**
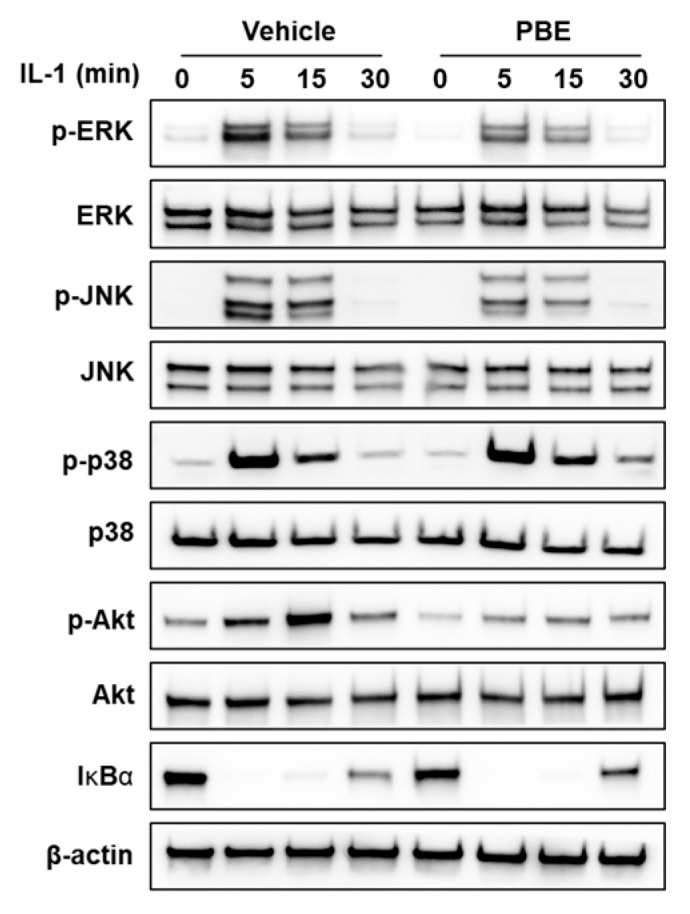
PBE attenuates early IL-1–triggered signaling in MLO-Y4 cells. MLO-Y4 cells were pretreated with vehicle or PBE (200 μg/mL) for 24 h and then stimulated with IL-1 (10 ng/mL) for the indicated times. Whole-cell lysates were analyzed by Western blotting to assess phosphorylation of ERK, JNK, and p38 MAPKs and Akt, together with IκBα degradation.

**Figure 4 nutrients-17-03273-f004:**
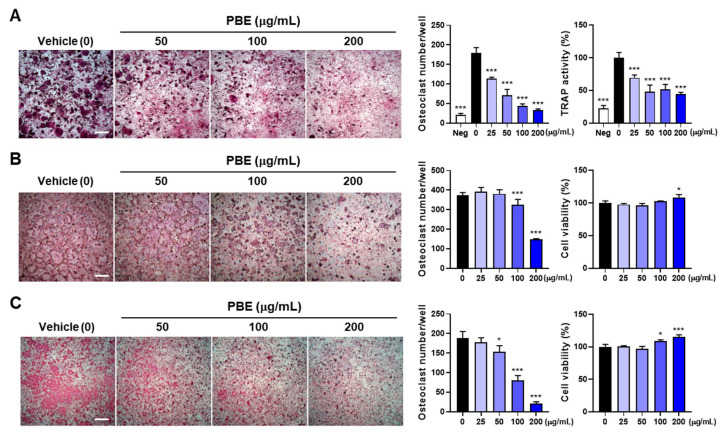
PBE inhibits osteoclast precursor differentiation in BMM cultures. (**A**) MLO-Y4/bone marrow cell co-cultures were pretreated with vehicle or PBE and then treated with PBS (Neg) or IL-1 (10 ng/mL) plus RANKL (50 ng/mL) for 5 days. Total TRAP activity and osteoclast number were quantified. (**B**) BMMs were cultured with M-CSF (60 ng/mL), pretreated with vehicle (0) or PBE, and stimulated with RANKL (50 ng/mL). Cell viability was assessed after 2 days, and TRAP activity and osteoclast number were measured after 4 days. (**C**) BMMs were treated as in (**B**), except TNF-α (20 ng/mL) plus TGF-β1 (1 ng/mL) replaced RANKL; viability was measured at 2 days and osteoclastogenesis at 5 days. Scale bars, 500 μm. Data are presented as mean ± SD (*n* = 3).* *p* < 0.05, *** *p* < 0.001 versus vehicle.

**Figure 5 nutrients-17-03273-f005:**
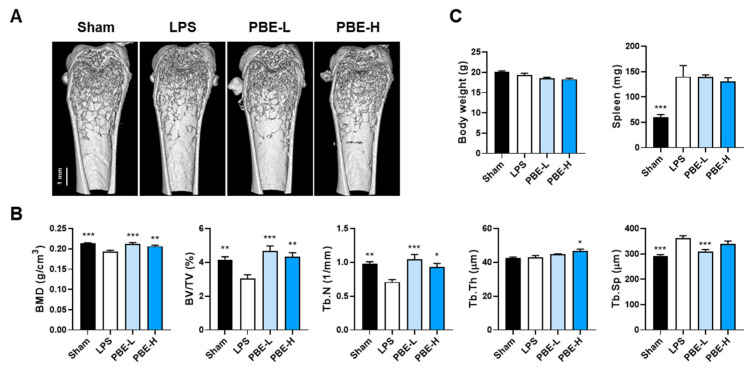
PBE protects against LPS-induced trabecular bone loss in mice. Mice received intraperitoneal injections of PBS (sham) or LPS (5 mg/kg) on days 0 and 4 and were sacrificed on day 8. PBE (100 mg/kg, PBE-L; 300 mg/kg, PBE-H) was administered by oral gavage once daily from 1 week before the first LPS injection until sacrifice. (**A**) Representative three-dimensional μ-CT reconstructions of distal femora (scale bar, 1 mm). (**B**) Quantitative trabecular morphometry, including BMD, BV/TV, Tb.N, Tb.Th, and Tb.Sp. (**C**) Body weight and spleen weight. Data are presented as mean ± SEM (*n* = 7). * *p* < 0.05, ** *p* < 0.01, *** *p* < 0.001 versus LPS control.

**Figure 6 nutrients-17-03273-f006:**
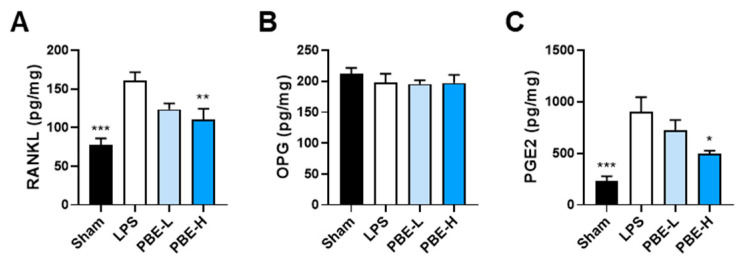
PBE reduces RANKL and PGE2 in bone tissue from LPS-challenged mice. Femoral bone lysates from the animals in Figure 5 were analyzed by ELISA for RANKL (**A**), OPG (**B**), and PGE2 (**C**). Data are presented as mean ± SEM (*n* = 6–7/group). * *p* < 0.05, ** *p* < 0.01, *** *p* < 0.001 versus LPS control.

## Data Availability

The original contributions presented in this study are included in the article. Further inquiries can be directed to the corresponding author.

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
