# Peer review of "Protaetia brevitarsis seulensis Larvae Extract Attenuates Inflammatory Osteoclast Differentiation and Bone Loss"

_nutrients, 2025, doi:10.3390/nu17203273_

Round 1
Reviewer 1 Report
Comments and Suggestions for Authors
Reviewer’s comments
This study investigated the inhibitory effect of the larvae of Protaetia brevitarsis seulensis larvae (PB), an edible insect, on osteoclast differentiation in in vitro co-cultures of mouse skeletal muscle and MLO-Y4 cells. It was found that PB inhibits osteoclast formation. Furthermore, the effect of inflammatory lipopolysaccharide (LPS) administration on bone loss in mice was examined, revealing that PB extract administration prevented bone loss. The results suggest that PB extract is effective in preventing bone loss. This study provides valuable new insights for the field of nutrition.
The reviewer's specific comments are listed below.
Introduction:
The nutritional components of PB larvae and their extracts have been described. It is important to note whether they contain any specific chemical factors.
Experimental method:
- An explanation of MLO-Y4 cells is required.
- In cell culture experiments, the number of cells used must be specified.
- The amount of protein used in western blotting analysis must be described must be described.
- In animal studies, the amount of protein used in Western blotting analysis must be described.
- In ELISA experiments, the use of cell culture medium must be described.
- Detailed descriptions of co-culture experiments and in vivo bone marrow cell cultures are required. The number of cells used must be described.
Results
In cell culture experiments, findings on whether the effects of PBE on cell number during culture must be described. If PBE reduces the number of cells in culture, this may affect the experimental results.
Discussion
This study used PBE extracted with 70% ethanol. The nutritional factors contained in 70% ethanol extracts are limited, particularly those containing highly polar chemicals. If findings on the factors contained in PBE extracted with 70% ethanol have been reported, it is important to describe that information.
Reviewer 2 Report
Comments and Suggestions for Authors
The manuscript entitled “Protaetia brevitarsis seulensis larvae extract attenuates inflammatory osteoclast differentiation and bone loss” presents a well-organized study investigating the anti-osteoclastogenic and anti-inflammatory effects of Protaetia brevitarsis seulensis larvae extract (PBE). The experimental design integrates in vitro and in vivo. The manuscript is generally well written, with strong methodological rigor and logical data interpretation.
ABSTRACT
The abstract is clear but overly detailed in methods; condense and emphasize key findings and potential implications.
MANUSCRIPT
- While the study demonstrates inhibition of ERK, JNK, and Akt phosphorylation, the direct molecular targets of PBE remain unclear. Please, try a possible explanation
- The choice of 100 and 300 mg/kg/day for in vivo experiments lacks justification. Provide a dose rationale based on prior toxicity, pharmacokinetic, or efficacy studies. If unavailable, acknowledge this as a limitation.
- The LPS-induced bone loss model represents an acute inflammatory state. Discuss how these results might translate (or not) to chronic inflammatory bone disorders (e.g., rheumatoid arthritis, postmenopausal osteoporosis).
- Since PBE is proposed as a “food-derived candidate,” discuss whether its oral bioavailability or metabolic stability supports potential nutraceutical use. Provide any supporting evidence or acknowledge the data gap.
MATERIALS AND METHODS
- Specify the purity of ethanol extract and whether batch variation was tested.
- Clarify storage duration and stability conditions for PBE.
ENGLISH
Several sentences are lengthy and could be shortened for readability, especially in the Introduction (lines 42–89).
Round 2
Reviewer 2 Report
Comments and Suggestions for Authors
The authors addressed all my comments; therefore, the manuscript can be accepted.